# Endangered Exotic Pets on Social Media in the Middle East: Presence and Impact

**DOI:** 10.3390/ani9080480

**Published:** 2019-07-24

**Authors:** Leonarda B. Spee, Susan J. Hazel, Eleonora Dal Grande, Wayne S.J. Boardman, Anne-Lise Chaber

**Affiliations:** School of Animal and Veterinary Sciences, The University of Adelaide, Mudla Wirra Road, Roseworthy, SA 5371, Australia

**Keywords:** Exotic pet trade, digital epidemiology, online influence

## Abstract

**Simple Summary:**

The exotic pet trade is impacted by social media via greater accessibility to photos and videos including these species and the increasing popularity of online animal marketplaces. The social media presence of public figures owning exotic pets has a major influence on exotic species demand. This study aimed to investigate exotic pet popularity, featured species on social media in the Middle East, and public perception of the animals spotlighted by public figures. We discuss the impact of such on the exotic pet trade and possible solutions to this issue. Male public figures from the United Arab Emirates predominated in the collected data, with most posts sourced from Instagram^®^. Eighty-five percent of the species displayed on social media posts were Convention on International Trade in Endangered Species of Wild Fauna and Flora (CITES) Appendix I- and II-listed, including big cats, birds of prey, and great apes. Through an investigation of social media posts of public figures in the Middle East, we found that there was an overall positive audience reception toward endangered exotic pets. Geographic region, social media platform, animal species, and animal age all influenced the results. We recommend improving public education and awareness of wildlife conservation and laws regarding exotic pet possession to combat the idolization of the exotic pet industry.

**Abstract:**

The popularization of exotic pets on celebrity social media in the Middle East has led to questionable impacts on exotic pet demand and threats to species conservation. The objective of this study was to identify exotic animal species featured on Middle Eastern celebrity social media account posts, the public perception of those posts, and their potential impacts on exotic pet demand and conservation (for global-scale extrapolation). Public social media accounts of highly influential persons from oil-rich Middle Eastern regions were manually investigated to evaluate subject demographics, species features, and post information (likes, comments) between January 2017 and August 2018. Twenty-five subjects possessed active social media accounts, from which 418 social media posts were extracted based on their inclusion of a privately owned exotic animal. SPSS Version 25 was used for frequency and descriptive analyses of these posts, in addition to comment analyses to evaluate quantitative (emojis) and qualitative (text) audience perceptions from a total of 10 social media posts of CITES Appendix I- or II-listed species. A greater frequency of positive than negative comments was observed (*n* = 8017), demonstrating the higher likelihood of social media promotion rather than negation of the exotic pet trade. Public education on wildlife conservation and exotic animal trade risks is imperative for successful conservation and welfare protection.

## 1. Introduction

Exotic pets associated with high-profile celebrities are portrayed as lavish contributions to already extravagant lifestyles. For this paper, exotic pets are defined as species originating from foreign regions that are not commonly popularized as companion animals and that lack an extensive history of domestication in a given location [1]. In this study, celebrities are defined as royal family members of Middle Eastern regions who maintain a prominent status in society and an influential impact over their reigning regions. While pop culture is an ever-evolving subject, royal family presence in these regions does not wane with trend changes, allowing for the maintained relevance of this group over time, resulting in a positive impact on the public influence of these figures. The public reach of celebrities has grown since the rising popularity of social media platforms such as Instagram^®^, Twitter^®^, Facebook^®^, and YouTube^®^. The ever-growing userbase of these platforms increases access for influential figures to their followers, with platforms such as Twitter^®^ having average gains of 135,000 users per day worldwide [2]. As of June 2018, Instagram^®^ has approximately 1 billion active users monthly, making the potential impact of celebrity promotion of exotic pets via posted images and videos on social media platforms a major concern for the growth and maintenance of the exotic animal trade [3]. The most popular celebrity involved in this study possessed a reach of over double the total population of Dubai in August 2018, with 6.4 million followers on Instagram™.

The illegal trade of exotic species has an estimated global worth of USD 20 billion [4]. Minimal regulation and monitoring of illegal animal movement poses risks for animal welfare and conservation. These exotic species may consequently reach the pet, fashion, and tourism industries under inappropriate conditions unfit for their survival and well-being. In the United Arab Emirates (UAE), birds, reptiles, invertebrates, mammals, and fish are the most popular taxa involved in the exotic pet trade [5]. Increased interest in exotic animal possession and trade may result from influences seen on social media, similar to fluctuations in dog breed popularity based on media coverage [6].

The identification of exotic pet species included in social media posts of celebrities in the Middle East can be used to gauge the way in which the public perceives exotic animals in this format. Considering this information, the profound impact of this on public demand for exotic pets and the influence of such on conservation and wildlife trade may be investigated. The increased prevalence of social media reflects the widespread use of emojis to convey the emotions and thoughts of individuals, and thus the role of modern communication posed an impact on this investigation, unlike studies that do not consider this form of language. Additionally, the international colloquialism of emojis assisted in global-scale extrapolation of what these reactions to exotic animals on social media convey from the public, due to the worldwide application of emojis in everyday life. The detriment to local communities, as well as global-scale effects involving animal welfare, disease transmission, and species invasion into naive ecosystems, are just a few factors that are implicated in the exotic pet trade.

The aims of the study were to investigate the presence of exotic animals on celebrity social media accounts in the Middle East, their featured exotic species, their conservation status, and the public response to these social media posts.

## 2. Materials and Methods

### 2.1. Online Data Collection

Major regions of the Middle East were identified to define the parameters of the study, including all states of the UAE (Abu Dhabi, Dubai, Sharjah, Ajman, Umm al-Quwain, Fujairah, Ras al-Khaimah), Bahrain, Qatar, Saudi Arabia, Kuwait, Oman, and Yemen (Figure 1). Limited public social media activity from regions including Kuwait, Oman, and Yemen restricted information collection. Regions not included in the investigation due to the absence of royal families or the presence of turmoil that would interfere with data collection were Cyprus, Egypt, Iran, Iraq, Israel, Jordan, Lebanon, Palestine, Syria, and Turkey. Subjects were selected on the basis that they were the leader, a partner of the leader, or a child of the leader of an assessed region (i.e., a member of the royal family). Royal family members were chosen on the basis of popularity regularity. Although trends may vary and celebrity relevance to the public eye may vary over time, the reach of royal families across a range of age and gender demographics may be maintained due to their high-standing role in these societies.

The four social media platforms investigated were Instagram^®^, Twitter^®^, Facebook^®^, and YouTube^®^. Publicly available social media accounts of each subject were manually found via search engine (Google^®^) using the name of each subject. No social media posts uploaded prior to January 2017 were included in the study. Inclusion criteria for this portion of the study was the presence of an exotic animal kept in a domestic setting (i.e., an exotic pet) that was included in a social media post. Each post was assessed, extracted, and recorded into Microsoft Excel based on the following categories: Country, state, name of subject, gender of subject, age of subject, social media platform, number of followers, species, media form (image or video), age of animal (juvenile or adult), caption (Arabic), caption (English translation), like count, comment count, number of views, post web address (URL), date posted, date assessed, and number of days between assessed and posted dates. Subject information such as age, country, and gender was sourced via external online research. The number of followers of a subject, like count, comment count, and number of views for each collected social media post was recorded at the time of collection. Considering this, the age of a social media post ranged from 0–1.5 years, depending on the date on which each post was originally uploaded to a respective social media platform, which may have influenced the exposure and popularity of a given post. This resulted in 418 posts from 146 subjects. At the end of the data collection period, a sample of comments was collected based on posts with the greatest number of likes per species. The post with the maximum number of likes was selected per species for comment sampling. Comment collection was extracted manually, working backward from the most recent comment. Collection for posts with high comment volumes (e.g., thousands) was ceased once the compiled comments for each post per species reached 30 pages worth of collected data. This was due to the overabundance of comments to assess as well as to avoid overrepresentation of more popular species as much as possible. Due to the high number of comments for some posts, it was not possible to access initial comment entries, and thus accurate comment sampling of the entire collection of comments on each social media post could not be completed. The assessed species were chosen according to their presence on Convention on International Trade in Endangered Species of Wild Fauna and Flora (CITES) Appendix I- or II-listed species due to their conservation status and highly portrayed role in the exotic animal trade. In this case, the CITES listing for the species noted in social media posts was considered, but was not determined as a specific inclusion criterion. Species were chosen for further comment analyses on the basis of more commonly known or popularized species on the global platform.

### 2.2. Data Consolidation

Comments were categorized into broad terms (p, positive; o, neutral; n, negative) and expansive terms (a, angry; c, animal; h, happy; m, miscellaneous; r, geographic region; s, sad; w, shocked; Table 1). The sentiment behind these terms and emojis (i.e., whether a given term was either positive, negative, or neutral) was decided by the conductors of the study prior to commencement of comment analyses. A complete list of emojis encountered in the study is listed in Appendix A. Categorization was based on emoji classification, where the most commonly used emojis in the sampled comments were divided into each of the described subcategories. Comments were stripped of hyperlinks, and the number of emojis in each subcategory were summarized using the R Version 3.3.2 program and SPSS Version 25.

### 2.3. Data Analysis

Data management was undertaken in R Version 3.3.2. Descriptive analyses of the data collected from the social media posts, including frequencies, mean, median, and chi-square tests of information concerning the involved species and subject demographics, were conducted in SPSS Version 25. A total of 418 social media posts were included in the scope of the study from 146 subjects. Social media posts made by a single highly active subject (responsible for 297 of the total 418 social media posts) were removed to reduce the effect of bias due to data skewing. The resulting adjusted data was comprised of 121 social media posts after removal of this subject. Although this highly active subject was removed for analytical reasons, for the purpose of this study the total and adjusted datasets will not be compared throughout the results. The following categories were recoded (i.e., combined) due to an insufficient sample size in some categories (i.e., *n* < 5): Region (UAE, non-UAE), species groups (mammals excluding primates, primates, birds, reptiles), social media platform (Instagram^®^, non-Instagram^®^).

A maximum of 30 pages of comments were collected for each analysed species group for comment data analyses. Thus, it is important to note potential differences in actual total comment volume. Emojis (illustrated icons used to demonstrate an emotion or reaction to online content) were prevalent throughout extracted comments and were categorized using R (Appendix A) based on work undertaken elsewhere [7]. The categorization of comments based on emojis was carried out for broad categories (positive, neutral, negative) and expansive categories (angry, animal, happy, love, miscellaneous, geographic region, sad, shocked). R reformatting was utilized for qualitative comment analysis and analysed using Excel to highlight key positive (happy, cute, love, funny, haha, lol, hehe) and negative (bad, stop, poor, sad, hate, cry, angry) phrases in comments. Key phrases were categorized as either positive (p) or negative (n). Positive perspective search terms included cute, love, happy, and yes. Negative perspective search terms included angry, sad, upset, and no.

## 3. Results

### 3.1. Social Media Posts

#### 3.1.1. Frequency Analysis

Of the total assessed subjects, 37% (54/146) of the celebrities possessed at least one public social media account, and 46.3% (25/146) of the celebrities with social media accounts featured exotic animal species in at least one post during the study period. The majority of social media posts from all social media platforms were sourced from the UAE (86.1%, 360/418), followed by Bahrain (10.8%, 45/418), Qatar (2.6%, 11/418), and Kuwait (0.5%, 2/418). Dubai was the most active state on social media (85.2%, 356/418), followed by Bahrain (10.8%, 45/418), Qatar (2.6%, 11/418), Ajman (0.7%, 3/418), Kuwait (0.5%, 2/418), and Abu Dhabi (0.2%, 1/418). It must be noted that Dubai, Ajman, and Abu Dhabi are all states of the UAE. Instagram^®^ (Facebook, Inc., Cambridge, MA, USA) had the highest number of posts compared to the other social media platforms examined in this study (89.5%, 374/418), followed by Twitter^®^ (5.3%, 22/418; Twitter, Inc., San Francisco, CA, United States, USA), Facebook^®^ (4.1%, 17/418; Facebook, Inc., Cambridge, MA, USA), and YouTube^®^ (1.2%, 5/418; YouTube, Inc., San Bruno, CA, United States, USA). Animal species publicized via social media posts showed a broad range in popularity (Table 2). Video (60.5%, 253/418) predominated over image (39.5%, 165/418) forms of media. Adult exotic animals (68.2%, 285/418) were seen more often than juvenile exotic animals (31.8%, 133/418) on the analysed social media accounts. Mammals and birds were the most common taxa featured in social media posts.

#### 3.1.2. Descriptive Analysis

There was missing data on subjects for age (*n* = 2) and views (*n* = 190) due to a lack of disclosure on the public social media profiles of specific subjects. Views (*n* = 228) were available only from Instagram^®^ and YouTube^®^ posts, and thus this variable did not represent the entire social media platform category. A highly active subject was removed to minimize bias in the sample size for age, followers, likes, comments, and views variables (Table 3).

#### 3.1.3. Cross-Tabulation 

Cross-tabulation analysis was utilized to evaluate the relationship between variables with potential influence in collected social media posts. The social media platform was found to be associated with region (*p* < 0.001), gender (*p* = 0.012), and exotic animal species (*p* = 0.010) (Appendix A). Exotic animal species was found be associated with region (*p* < 0.001), gender (*p* < 0.001), animal age (*p* < 0.001), and media form (*p* = 0.008) (Appendix A). An association was found between animal age and region (*p* = 0.016) (Appendix A).

### 3.2. Comment Analysis

#### 3.2.1. Quantitative Analysis 

A total of 8017 comments were analysed among the 10 species. The number of pages of comments for analysed posts of each species group were as follows: Arabian oryx (395 pages), giraffe (253 pages), falcon (220 pages), elephant (76 pages), cheetah (6 pages), orangutan (5 pages), chimpanzee (4 pages), tiger (3 pages), and zebra (2 pages). The descriptive statistics for the distribution of emojis per comment are represented via condensed and expansive categories (Table 4).

Overall, 4.1% of comments included purely negative emojis, 46.9% of comments included only positive emojis, and 3.6% of comments included both positive and negative emojis simultaneously. Neutral comments including neither positive nor negative key words or emojis comprised the remaining fraction of comments.

#### 3.2.2. Qualitative Analysis

A frequency table of key phrases was evaluated in the data (Table 5). Based on the assessed key words, the majority of text comments were positive (86.3%). Comments including negative key words comprised 13.7% of the analysed comments. 

## 4. Discussion

The aims of the study were to investigate the presence of exotic animals on celebrity social media accounts in the Middle East, featured exotic species, and their conservation status, in addition to the public response to these social media posts. Of the assessed celebrities who possessed public social media accounts, almost half of the subjects presented exotic pets in one or more posts between February 2017 and August 2018. Considering this, the majority of these posts were found on Instagram^®^, with this being the most popular social media platform in the findings. The exotic species most likely to be featured by celebrities were falcons, giraffes, the Arabian oryx, and gorillas (adjusted data; Appendix A). CITES and the International Union for Conservation of Nature (IUCN) categorizations were used to evaluate the conservation status of exotic species observed in collected social media posts. An overall positive public response to these social media posts was found for both quantitative (emojis) and qualitative (keywords) comment evaluation.

### 4.1. Celebrities in Middle Eastern Countries

This study evaluated social media activity of celebrities across the Middle East, where an investigation was plausible. The presence or absence of exotic animals on these social media pages may not accurately reflect the true prevalence of exotic animal ownership. As each subject has a choice in which animals are publicized on social media, not all exotic animals owned by or associated with celebrities in the Middle East are included in this data. It must be considered that there is no official data on exotic pet ownership for a large portion of cases in this region.

#### 4.1.1. Gender

There is an inherent difference between traditional media and social media considering accessibility, and the ability to reach and directly interact with followers or members of the public through social media. The depiction of males as more active on social media with regard to exotic species-related content (78.5%) compared to females (21.5%) in the data implied a contrast in audience influence. Females on social media have a greater reach toward their public audience with greater levels of likes and share popularity compared to males, as found in a study based on Israeli social media users [8]. There was nevertheless no association found between gender and number of likes or number of comments in our study. During data collection, it was noted that the majority of social media posts, including exotic species, were sourced from male subject social media accounts (not considering the removed highly active subject). This perhaps indicated a greater level of exotic species ownership by male subjects.: However, this perception may have been influenced by the greater male social media presence relative to female subjects included in this study.

#### 4.1.2. Social Media Platforms

In 2017, Instagram^®^ (the most popular social media platform in our study) introduced a filter system to notify users of potential breaches of animal welfare based on their searches on the app in order to promote awareness of wildlife exploitation [9,10]. Upon searching hashtags such as “#slothselfie” or “#elephantride”, users are shown a pop-up notification stating Instagram^®^’s negative stance on animal abuse and the endangered animal trade.

Despite efforts by social media platforms, installed protection systems are unable to filter all posts relating to wildlife exploitation. The accumulation of exotic animals displayed online as pets or in captive settings may consequently impact wildlife exploitation and welfare via customer attraction [11,12]. Slow lorises are an endangered primate species group frequently seen online as props in tourist attractions. As a result of popularity gained on Instagram^®^, slow lorises have been used for this purpose in a wide geographical range from Thailand to Europe, shown in poor welfare situations with an inappropriate diet and exaggerated light exposure [11]. This expansiveness displays the proliferation of trade for this species, which potentially extends to a proliferating wildlife trade of other endangered species due to social media popularity. Personal user accounts may be used for the illegal trade of exotic animals. In 2015, the Cheetah Conservation Fund (CCF) uncovered 369 accounts on Instagram^®^ advertising the sale of live cheetahs [13]. Evidence of the exotic wildlife trade of both dead and living animals has also been found on Facebook^®^, with over 700 advertisements for a range of exotic species identified in 2016.: However, trade monitoring is limited due to the private nature of these sales [14,15].

### 4.2. Public Reception of Social Media Posts

This display of wildlife on social media in common settings can impact the public perception of species’ statuses in the wild. It has been found that the more frequent presence of threatened species on social media platforms is associated with the perception of increased normalcy of these species and decreased concern for their vulnerable status [11]. This was reflected in the more positive response by the public toward exotic animals kept as pets, which was seen in the results.

Audience perception of different emojis is influenced by factors such as geographical location, gender, and the operating system of the device upon which the emojis are viewed [16]. The use of emojis has been associated with enhanced happiness levels in a comment’s author [16]. This was reflected by the lower negative emoji prevalence compared to positive emojis in the comment samples. As comments were assessed based on emojis, the absence of negatives could have skewed the data, thus potentially indicating a greater positive response to exotic animals on social media than is accurate. The inclusion of emojis in comments related to serious topics has been associated with lower levels of interest in these topics by the writer [17]. This indicates that negative comments where users are trying to convey serious opinions on the topic of exotic pets may be hidden in this analysis due to an absence of emojis.

Approximately 3.6% of the comments included both positively and negatively classified emojis. Sad emojis (negative broad classification) generally do not have a major impact on the perception of an adjoining piece of text [18]. However, it must also be noted that positive emojis are commonly used with “ironic” and “polite” connotations as well as for direct positive meanings [19]. Furthermore, there was the additional potential that the attitude of public comments may not have reflected the study objective. A comment may represent the views of a user based on the popularity status of the subject or the featured species itself and the appeal of the animal rather than a direct reflection of the user’s opinion on the exotic animal trade. Therefore, it is difficult to determine the true intent of the chosen emojis due to the variation in perceptions of these emojis by social media users. As this study investigated only social media posts regarding exotic species-related content, the public reception to these posts based on this data cannot be compared to posts not associated with this area.

There were 72.6% more positive comments based on the presence of positive keywords compared to negative comments containing negative keywords. Language barriers, typos, and spelling variations undetected during the analysis may have resulted in this difference. There was the potential that keywords may have been used for ironic or non-direct purposes, as seen with positive emojis [19]. This could not be detected via the methods of analysis used in this study.

Potential bias was a possibility in subjects who deleted comments deemed unfavourable to their image, which would have been reflected in a lowered negative response toward their exotic animal posts. As comments were extracted from the most popular social media posts based on like counts, non-uniformity was present in the total number of analysed comments per species. This may have produced a skew in analysis results caused by varied sample sizes. Furthermore, the popularity of an individual subject may have influenced the common theme of comments by the public on their social media posts. An overall positive or negative comment tone may have been a direct reflection of an individual’s popularity.

### 4.3. Social Media and Population Trends in the Middle East

In January 2018, a 32% annual growth of social media users was reported in Saudi Arabia (10% greater than the worldwide average), indicating an increased demand for social media and its products [20]. Middle Eastern countries including Jordan and Lebanon, although not looked into greatly in this study, also produced the highest volume of social media use in 2018, with a spike seen in this statistic during 2015–2017, which may have radiated across other Middle Eastern countries [20]. An influx of users in these areas of the Middle East was reflective of a boom in population, resulting in an increase in the social media-savvy demographic, with more than 60% of Middle Eastern citizens being under the age of 30 in 2015 [21]. The elevated growth of male relative to female populations due to an imbalanced gender ratio (51:49) was reflected in data found in a study on gender and social media use (however, in this case, it must be considered that only exotic species-related content was considered for the study data) [21]. Sixty-three percent of the younger population in the Middle East has been reported to use social media (Facebook^®^™, Twitter^®^™) as a primary source of news and information, rather than traditional media sources such as television and official online news platforms [20]. This dependence on social media further supports the influence social media has over online users in this region. Despite there being a clear division between the socioeconomic status of members of Middle Eastern monarchies and the citizens of these countries, the social environment between these groups remains respectful and in favour of nation rulers and their families [22]. This would explain the large following of these rulers on social media and the largely positive response toward their lavish lifestyles from nonmonarchs. Idolism toward these figures and their actions translates to a greater impact on their followers, thus perpetuating the popularity and acceptance of exotic pets in common settings. Between 2013 and 2014, approximately half of surveyed exotic pet shops were reported to sell CITES Appendix I-listed species (A-L Chaber, personal communication). This may have correlated with the predominant featuring of CITES Appendix I species in celebrity social media posts and the consequential impact of such on private purchases of exotic pets in the Middle East (Table 2).

### 4.4. Exotic Animal Trade 

The Convention of International Trade in Endangered Species of Wild Fauna and Flora (CITES) is an international entity that regulates the trade and movement of endangered or at-risk species [23]. All regions investigated in this study are included in this conservation effort (UAE, Qatar, Bahrain, Kuwait). CITES lists are divided into three categories of descending severity: Appendix I (heavily restricted trade), Appendix II (trade only allowed without threat to species survival), and Appendix III (trade allowed with export permit) [24]. A number of species included in the data are found on CITES lists, including but not limited to African grey parrots (Appendix I), cheetahs (Appendix I), and gorillas (Appendix I) (Table 2) [25]. Due to the at-risk status of species included on CITES lists, these animals are considered rarities and are considered to be more profitable in the illegal wildlife trade [26].

A highly active subject included in the total data of the study was removed from the adjusted data due to a potential representation of bias, which would have over-represented and skewed the study results. However, this subject should not be considered a true factor of bias. The high volume of posts focused on exotic pets kept by the subject rather than political commentary contrasted to many other subjects of the study. The subject promoted a number of species (e.g., big cats, great apes) that were expected to have a large focus in the study but that were largely not represented in the final collected data. This may suggest that the subject could act as an unfiltered view into the exotic pet trade and exotic pet ownership in the Middle East.

One of the most commonly noted species in the data was falcons (31.1%). This popularity in the UAE was mirrored in neighbouring regions, including Lebanon, Saudi Arabia, and Jordan [26]. In 2006, the UAE was found to be a main importer region of birds, with two main trade routes from Africa (parrots) and Europe (falcons) [27]. Illegal bird imports have historically caused major issues in the Middle East, such as in the 2005 H5N1 outbreak in Saudi Arabia due to the unapproved import of falcons from Mongolia [28]. Birds (specifically parrot species) and reptiles in private collections are more likely to be taken from the wild or sourced from first-generation captive parents, which may have a greater effect on wild populations of popular species [27,29]. However, it must be noted that birds of prey are commonly involved in now-legal captive breeding in most regions of the Middle East [29]. The Middle East was the largest importer and third largest exporter of mammals as of 2014 [26]. A matter to consider is the established father population of the Arabian oryx in the Middle East, which has resulted in the growth of the population of this species in the region due to natural breeding rather than imports (A-L Chaber, unpublished data). Over 50% of this population is located in the UAE [30]. Not all information on the mammal trade may be accurate. Globally, 46% of reported trade species are comprised of mammals, based on recorded data not including all exotic mammal trade forums [31]. Markets and private sales of individual exotic animals pass under the radar, which may impact statistics on the ownership and movement of these mammals internationally and in the Middle East [26].

### 4.5. Conservation Status of Featured Species

A recent decline in exotic species populations may be attributed to a number of causes, including the exotic pet trade, increased demand for these species due to tourism and fashion, as well as the normalization of the ownership of these species due to a drive in social media.

Tigers, the most commonly featured species in the data, experienced an approximate population loss of 55% over the past 20 years, which was mirrored by lions (54% loss), giraffes (38% loss), and gorillas (60% loss) over the last 30 years, as well as cheetahs (30% loss) over the past 15 years [32]. Considering this, it must be noted that the decline in these species may not have been directly impacted by social media influence in all cases, where other drivers of the exotic animal trade may have come into play. A topic that should be discussed, however, is the potential influence of social media on these outside issues propagating the popularity of exotic pet ownership and thus the exotic animal trade. In effect, species conservation is vulnerable to the effect of such influencing events.

The requirements to breed a number of species (e.g., cheetah) in captivity may surpass the knowledge and resources or conditions available to those wanting an exotic pet. Thus, in many cases, capture from the wild is the most fruitful option in acquiring an animal of a given species (13,27]. Up to 64.6% of live traded carnivores are sourced from the wild rather than bred in captivity [33]. As a result of this, there may be a greater impact on the conservation status of species unable to be bred in captivity. Discrepancies have been found between grey literature and CITES trade database recordings of the exotic animal trade, which may support the potential for unregulated trade and an unknown detrimental impact on conservation [27]. In some cases, conservation concerns may be greater according to the region in which a species is traded. The UAE, for example, has had two separate Secretariat and Standing Committee-recommended suspensions in the trade of species listed under CITES (1985, 2001) due to the free illegal trade of CITES-listed species and noncooperation toward these issues in trade matters [34]. Compliance from local authorities plays an integral role in the monitoring and control of the exotic animal trade both regionally and internationally, thus also having a major impact on the effect of species conservation status.

### 4.6. Law and Public Education

In terms of the presence of exotic species in the collected data in this study, it was assumed that the animals included in the social media posts were owned privately by the subjects themselves or that the animals were accessed via a private establishment that owned the animals. Due to the dominance of data sourced from the UAE, the laws regarding the exotic animal trade examined in this paper focus on UAE regions. Due to the recent introduction of new laws relating to the exotic animal trade industry in the Middle East, true statistics of current exotic animals in Middle Eastern private establishments cannot be sourced.

A gradual progression of changed laws on exotic animal ownership has passed through regions of the Middle East in recent times. In 2002, Federal Law No. 11 outlined regulations regarding the ownership and registration of the possession of exotic animals on private property [35]. Sharjah, a state of the UAE, is a major game-changer with regard to exotic animal ownership laws, with regulations against the ownership and breeding of dangerous animals that were introduced in 2014 having since been escalated to being introduced into UAE Federal Law on exotic animal trade and possession [13]. Ownership, the display of exotic animals in public settings, the trade of exotic animals, and the use of exotic animals for defence or as tools for attack are now considered offences punishable by large fines and possible imprisonment after the enactment of Federal Law No. 22 in 2016 [36,37]. Despite a ruling declaring that all exotic animals be surrendered to authorities by July 2017, this appears to not have applied to the subjects included in this study, who had social media posts that included exotic animals dated up to August 2018. This puts into question the efficacy of these laws on exotic animal regulation in the UAE over all citizens, as well as the methodology of private zoological collection assessment.

The grey area nature of the exotic animal industry causes great difficulty in the accurate tracking of the prevalence and regulation of the animal trade [29]. Although there are gaps in legalities in terms of species protection, ruling out legal methods in the wildlife trade also poses a potential risk for exotic species. The banning of legal trade may result in increased rates of illegal exotic animal trade with minimized regulation and monitoring, posing a greater harm to animal welfare and conservation in some species [30]. However, discussion regarding the effects of more strict trade regulations have found that a combination of this rule-out of both legal and illegal trade with improved education on conservation in communities involved in trade has beneficial long-term impacts on endangered exotic species involved in the wildlife trade [38].

Efforts by welfare groups such as the International Fund for Animal Welfare (IFAW) educate the public on the purpose of instigating laws on exotic animals in society [37]. A survey of university students in the UAE taken before and after an animal conservation talk detailed the effect of education on the perception and understanding of the effect of the illegal exotic pet trade. Prior to this talk, the majority of students perceived wildlife conservation as animals being held in captivity (A-L Chaber, personal communication). A 73% increase in students seeing the illegal trade of exotic animals as a negative effect on the wild populations of these species was seen after an information session on this topic (A-L Chaber, personal communication). In terms of the impact of education, a worldwide online survey found that legalities and zoonotic disease had a major influence on the public perception of exotic animal ownership, while species conservation and welfare had a minimal impact [37].

The influence of fashion and modern trends on societal values may play a pivotal role in the popularity of exotic pets and the exotic animal trade. With increased internet usage and exposure to exotic pets owned by influential figures and fashion brands online, the availability of these species has been found to increase over time. This has been noted in investigations that have placed countries such as Qatar, Bahrain, and the UAE in the top 10 rankings of involvement in the wildlife trade (based on increased internet usage between 2008 and 2013) [39]. The use of exotic pets as status symbols and features of wealth by celebrities and fashion labels may have a positive impact on the propagation of the demand for privately owned exotic pets and the wildlife trade. The glorification of ownership of these species is an area for investigation regarding the impact of such on the public perception of and demand for exotic pets. Further exploration into the possibility that the removal of such glorification may result in a decrease in consumer exotic pet demand is merited.

### 4.7. Digital Sociology and Epidemiology 

Digital sociology is the study of the influence of online media and how it is used in society, while digital epidemiology is a reflection of population trends as sourced via online mediums [40,41]. Both of these terms may be applied to the influence of social media on the exotic animal trade in the Middle East. Cultural dynamics are a major driving force of trending topics on social media. When a social media post becomes viral and increases in popularity, public perception and previously formed opinions on the topic are amplified [6]. For example, if a celebrity were to post a photograph of themselves posing with a lion on Instagram^®^, preformed opinions of lions in close-contact situations by an individual may result in either strongly negative or strongly positive opinions on the depicted situation. These opinions may influence the actions of an individual, whether it be to support or to reject the exotic animal trade through online activity surrounding this issue. The ability of social media platforms to portray exotic animals in everyday situations introduces a normalcy toward exotic animals kept in private facilities as pets [42]. As the public becomes accustomed to seeing this type of content online, the shock factor wears off and exotic animals become an accepted part of society, which is reflected in the prevalence of the exotic animal trade.

## 5. Conclusions

The UAE was found to have the greatest social media presence regarding the display of exotic species compared to the other evaluated Middle Eastern countries. Based on comment analyses from the investigated species groups, the public perception of celebrity exotic pets was overall positive. Despite increasing awareness of the consequences of the exotic animal trade via media outlets and government action in the Middle East, the exotic animal trade is a global issue. As a result, the presence of an international audience to these social media posts may have an impact beyond that seen in the Middle East. These findings indicate that more effective education on exotic animal protection in combination with recently improved laws and regulations on the subject is essential in order to combat the illegal wildlife trade, improve animal welfare, and increase conservation awareness in these countries.

## Figures and Tables

**Figure 1 animals-09-00480-f001:**
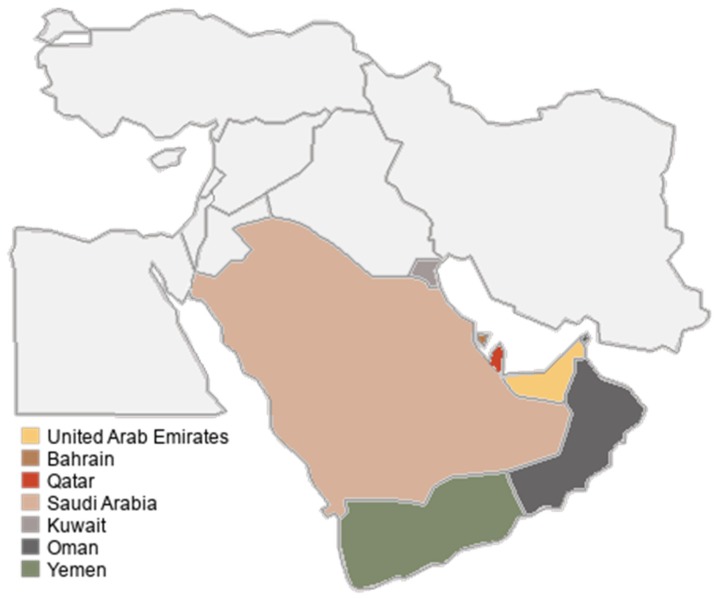
Map of geographical origins of Middle Eastern celebrities included in the social media activity analysis of exotic pet possession or involvement.

**Table 1 animals-09-00480-t001:** Sample of emojis evaluated under broad and expansive classifications in the extracted comments from social media posts of assessed species used to garner public reaction toward exotic pets featured on social media posts in the Middle East.

Emoji	Description	Classification (Broad)	Classification (Expansive)
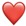	Red heart	Positive	Love
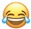	Face with tears of joy	Positive	Happy
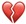	Broken heart	Negative	Sad
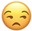	Unamused face	Negative	Anger
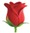	Rose	Neutral	Miscellaneous
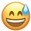	Smiling face with open mouth and cold sweat	Neutral	Shock
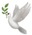	Dove	Neutral	Animal
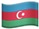	Azerbaijan (flag)	Neutral	Geographic region

**Table 2 animals-09-00480-t002:** Exotic animal species included in total collected data from social media platforms (Instagram^®^, Twitter^®^, Facebook^®^, YouTube^®^) of Middle Eastern celebrities.

Species	Frequency	Percent (%)	CITES Appendix ^1^	IUCN Red List Status ^2^
Tiger (*Panthera tigris*)	131	31.3	I	Endangered
Falcon (*Falco cherrug*)	130	31.1	II	Endangered
Lion (*Panthera leo*)	65	15.6	I	Vulnerable
Orangutan (*Pongo*)	31	7.4	I	Critically endangered
Giraffe (*Giraffa camelopardalis*)	12	2.9	-	Vulnerable
Arabian oryx (*Oryx leucoryx*)	10	2.4	I	Vulnerable
Chimpanzee (*Pan troglodytes*)	10	2.4	I	Endangered
Pharaoh eagle-owl (*Bubo ascalaphus*)	10	2.4	-	Least concern
African grey parrot (*Psittacus erithacus*)	3	0.7	I	Endangered
Gorilla (*Gorilla gorilla*)	3	0.7	I	Critically endangered
Cheetah (*Acinonyx jubatus*)	2	0.5	I	Vulnerable
African elephant (*Loxodonta Africana*)	2	0.5	I	Vulnerable
Exotic bird ^3^ (Psittacidae)	2	0.5	II	Family classification not possible
Coyote (*Canis latrans*)	1	0.2	-	Least concern
Flamingos (*Phoenicopteridae*)	1	0.2	II	Least concern
Ocelot (*Leopardus pardalis*)	1	0.2	I	Least concern
Peacock (*Pavo muticus*)	1	0.2	II	Endangered
Short-beaked common dolphin (*Delphinus delphis*)	1	0.2	-	Least concern
Tortoise (*Centrochelys sulcata*)	1	0.2	II	Vulnerable
Zebra (*Equus zebra*)	1	0.2	II	Vulnerable
Total	418	100		

^1^ Convention on International Trade in Endangered Species of Wild Fauna and Flora (CITES) Appendix entries listed as “-” indicates absence of entry on CITES Appendix I, II, or III; ^2^ International Union for Conservation of Nature (IUCN); ^3^ Specific bird species unable to be identified from social media post.

**Table 3 animals-09-00480-t003:** Descriptive analysis of total collected data, including (total) and excluding (adjusted) a highly active subject for measured variables: age, followers, likes, comments, and views.

Variable	*n* (Posts)		Minimum		Maximum		Median ($\tilde{x}$) Mean ($\bar{x}$)		Std. Deviation	
	Total	Adj.	Total	Adj.	Total	Adj.	Total	Adj.	Total	Adj.
Age	416	119	8	8	89	89	$\tilde{x}$ = 44.5 $\bar{x}$ = 31.24	$\tilde{x}$ = 44.5 $\bar{x}$ = 36.82	8.22	13.91
Followers	418	121	2113	2113	6,400,000	6,400,000	$\tilde{x}$ = 3,200,000 $\bar{x}$ = 591,287.34	$\tilde{x}$ = 3,200,000 $\bar{x}$ = 1,512,447.19	1,413,499.08	2,395,577.63
Likes	418	121	0	0	271,517	271,517	$\tilde{x}$ = 135,758.5 $\bar{x}$ = 12,569.42	$\tilde{x}$ = 135,758.5 $\bar{x}$ = 39,804.45	42,325.41	71,912.79
Comments	418	121	0	0	24,082	24,082	$\tilde{x}$ = 12,041 $\bar{x}$ = 562.94	$\tilde{x}$ = 12,041 $\bar{x}$ = 1852.17	2281.10	3964.93
Views ^1^	228	15	1316	3991	969,476	969,476	$\tilde{x}$ = 484,738 $\bar{x}$ = 25,084.47	$\tilde{x}$ = 484,738 $\bar{x}$ = 292,417.67	110,363.59	339,242.44

^1^ Views available only for Instagram^®^ and YouTube^®^ post entries.

**Table 4 animals-09-00480-t004:** Descriptive analysis for emojis used in individual comments per exotic animal species post classified through condensed categories (p, positive; o, neutral; n, negative) and expansive categories (*n* = 8017).

Category	Minimum	Maximum	Mean	Std. Deviation
**Condensed Categories**				
Positive	0	82	0.51	3.37
Neutral	0	44	0.68	2.19
Negative	0	27	0.15	0.78
**Expansive Categories**				
Love	0	82	0.93	3.02
Miscellaneous	0	28	0.60	1.94
Geographic region	0	25	0.02	0.38
Happy	0	24	0.57	1.48
Shocked	0	14	0.05	0.44
Sad	0	13	0.03	0.32
Angry	0	11	0.07	0.43
Animal	0	10	0.06	0.44

**Table 5 animals-09-00480-t005:** Frequencies and percentages of positive (happy, cute, love, funny, haha, lol, hehe) and negative (bad, stop, poor, sad, hate, cry, angry) keywords extracted using Excel found in collected Instagram^®^ comments. Data regards total species rather than individual species.

Keyword	Frequency	Percentage of Total Evaluated Keywords (%)	CITES-Listed Species of Total Featured (%)		
			I	II	III
Love	227	39.5	70	30	-
Cute	142	24.7			
Happy	62	10.8			
Stop	34	5.9			
Haha	31	5.4			
Lol	24	4.2			
Bad	15	2.6			
Sad	12	2.1			
Funny	8	1.4			
Hate	6	1.0			
Cry	5	0.9			
Angry	4	0.7			
Poor	3	0.5			
Hehe	2	0.3			
	575	100			

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
