# Peer review of "Endangered Exotic Pets on Social Media in the Middle East: Presence and Impact"

_animals, 2019, doi:10.3390/ani9080480_

Round 1

Reviewer 1 Report

Animals 543184

Endangered exotic pets on social media in the Middle East: Presence and Impact

This study examines the possible effect of social media on pubic perceptions of exotic pets and associated drives towards exotic pet keeping habits, and raises the welfare implications associated with popularisation of private animal keeping. The study appears to me to adopt a novel and relevant approach to assessing certain influences on pet acquisition behaviour, and its messages make sense. The authors are open about the possible interpretive limitations of their study, but I think their conclusions stand. The text is well written, and the data valid. I have only a few minor suggestions for text refinements, which I include below.

Check spellings such as ‘analyse’ and ‘analyze’ throughout for consistency.

Line 44

Reads

Exotic pets are defined…”

Suggest reads

“For this paper exotic pets are defined…”

Line 221

Reads

“Upon”

Use ‘On’ unless referring to putting something ‘up and on’

Line 228

Reads

“Since”

Use ‘Because’ in place of ‘since’ unless referring to a timeframe – i.e. ‘since 1998’

Line 271

Reads

“in analysis results as result”

Needs rewording.

Line 307

Reads

“African Grey Parrots”

Animal names should read, e.g.

“African grey parrots”

Line 385

Reads

“The grey area nature of the exotic animal trade industry”

Suggest reads

‘The grey area nature of the exotic animal industry”

Lines 388-390

Reads

“Banning of legal trade would result in increased rates of illegal exotic animal trade with minimized regulation and monitoring, posing greater harm towards animal welfare and conservation.”

Comment

This conclusion is incorrect. Ample evidence exists to show that while bans may – possibly - cause minor sort term increases in illegal trade in some species, these measures (bans) are the most effective means of reducing or eliminating trade in exotic pets, and there are numerous references to support this position, e.g.:

- Reino et al., (2017) Networks of global bird invasion altered by regional trade ban. Sci. Adv.3:e1700783

- Toland et al., (2012). The exotic pet trade: pet hate. The Biologist, 59(3), 14-18.

Reviewer 2 Report

Introduction

Line 44: A more common definition of exotic pets is animals without a long history of domestication that are non-native to an area.

Lines 47 – 51: It would be more useful to give the number of followers that the celebrities have (maybe as a proportion of the population). The overall stats of the social media sites are impressive but unclear if it’s applicable to your study population.

Lines 54-62:

Introduction overall:

The intro is a good start to for background information. Here are some questions I am asking myself after reading it, which, when answered, may improve the manuscript:
•    What are you research aims or questions?

•    Why is your study important? I didn’t get the implication that your study might have. Increased understanding of public perception of exotic pets? Is this filling a research gap? What about other studies that have looked exotic pets on social media?

•    Why look at Middle Eastern celebrities? What’s different about them that makes them a useful case study? Do a large portion of the countries follow them? Are they revered?

•    What are the risks and impacts of the exotic pet trade? Said another way, who cares if these pets are traded? There’s room here to talk about conservation risks, invasive species, disease transmission, and welfare.

Methods

Line 71: Are there other celebrities in these countries other than royalty? If for the most part no, then this is a good rational for looking at them. If there are other celebrities with potentially more followers, why not look at them?

Line 78: Did you only recorded a post if it included an exotic animal? Is that your inclusion criteria?

Line 82: Why only take a sample if you have all the comments (too many I assume)?

Line 82: How did you sample comments (random, other)?

Lines 88-90: Did you only look at CITES listed species? Were the species that were not in CITES? This gets back to inclusion/exclusion criteria.

Line 92: Did you reference anything when categorizing emojis? See https://medium.com/analytics-vidhya/simplifying-social-media-sentiment-analysis-using-vader-in-python-f9e6ec6fc52f for previous look at extracting the ‘sentiment’ of emojies and text

Line 106: frequency, mean, median, chi squared of what exactly?

Line 108-110: This sentence doesn’t make sense to me. Did you remove 121 posts? Why?

Methods other:

•     You might want to look at IUCN status as well. I suspect that most of these are endangered. This would be a quick to do and would be a meaningful addition. https://www.iucnredlist.org/

Results
Table 2: looks like 2 tables are combined into 1 table. Could be a formatting error by the journal?
Table 3? – the table with the subject summary stats. I think histogram distributions would be more useful here than min, max, median, standard deviation. But that’s more of a personally preference.
Table 3 – what does adjusted mean? How did you adjust? Please mention in methods.
Table 3 – How did you get Age? When did you record followers, likes, comments, views? At the time of collection? How long after the post did you wait to record – this will affect the number of likes, comments, and views. Please mention in methods.
Line 156: Where is the bold and asterick data? I don’t know what you’re referencing to. Also what do you mean by cross tabulation?
Appendix B, C, D: I don’t find these analyses particular useful or informative. I think you don’t need them – the analyses in the main text get at what you’re trying to do.
Results other:
•    How can you be sure that the emoji reactions to the post are for the animals and not for the celebrities? You don’t have a baseline to compare to. For example, what if all non-animal posts have the same degree of ‘positive’ response? It looks like you didn’t collect this data, which is fine. You just need to discuss this and the limitations of the inferences you can draw from it.
•    The same thing when you look at gender. What is the gender ration of non-animal posts? Is it different than animal posts?

Discussion

Line 191-193: Nice! I think this sentence should also go in the last paragraph of the introduction.
First paragraph: worth mentioning that these are CITES listed species and IUCN endangered (if you  do that).

Section 4.1 and subsections: A lot of this seems like data summary and should be in the Results section. Discussion of results and implications should be reserved for discussion section. Also, a lot of it looks like it was repeated elsewhere in the manuscript – so not needed to repeat in the discussion.

Line 224: Is it against social media policy to post with animals though? I know it is to sell animals but I’m not aware of just posting pictures them.

Line 227: Yes good point. Also, worth mentioning in introduction as rationale to explore this.
Section 4.2: I would make sure to caveat that you didn’t compare the sentiment of non-animal posts.

Line 284: I thought this ratio was for animal posts not for all posts? If yes, then these needs to be rewritten with the actual ratio of all users or omit sentence.

Line 297: personal communication – it would be great if this was data was published so other researchers can use it. Also, it seems weird to be so specific with the numbers if the data isn’t published. Perhaps, half of all survey store sold CITES species?

Line 311: I don’t understand why this subject was omitted. It seems like it should be included in your analysis.

Line 335 – 339: With no evidence other than word of mouth I would omit this from the manuscript.

Line 341 – 350: I would rephrase to say there a multiple reasons for the decline of these species (list reasons). The exotic pet trade is one and social media may be increasing demand/normalize the keeping of these animals.

Sectiton 4.5, paragraph 2: seems like a long winded way to say there is illegal trade of these animals despite CITES listed.

Line 369: what ‘new laws’ are you referring to? Also, why are you talking about private ownership? I think more of an introduction into this sentence is needed.

Line 391 – 400: why not cite the study instead of personal communication? I would try to stick to primary literature as best as possible. The multiple uses of personal communications makes me a bit uneasy. The editor may feel differently.

Line 408 – how can we be sure that removing glorification over ownership will decrease demand? That seems like a reasonable hypothesis so I would reframe to say that this merits investigation.

Section 4.6 overall: I don’t really see how this section sits with your study. I think it can be reduced down to the main points – there are laws but not enforced + enforcing laws could help stop glorification which could decrease desirability of these animals as pets.

Discussion overall:

There are a lot of good elements here. Personally, I thought it was a little to disjointed to follow the discussion especially in relation to your study. I think with reworking and refining the text to distil down the important parts that pertain particularly to your study will make for a better discussion. Following the comments above are a good start. Personally I would get rid of the section headers and just write in a paragraph format.

Citation 41 – url doesn’t work
